# Designable Poly(methacrylic Acid)/Silver Cluster Ring Arrays as Reflectance Spectroscopy-Based Biosensors for Label-Free Plague Diagnosis

**DOI:** 10.3390/polym15081919

**Published:** 2023-04-17

**Authors:** Chih-Wei Chen, Shih-Hsun Chen, Chih-Feng Huang, Jem-Kun Chen

**Affiliations:** 1Division of Neurosurgery, Department of Surgery, Chi Mei Medical Center, Tainan 710, Taiwan; awei921@gmail.com; 2Department of Occupational Safety and Health, Institute of Industrial Safety and Disaster Prevention, College of Sustainable Environment, Chia Nan University of Pharmacy and Science, Tainan 717, Taiwan; 3Department of Materials and Science Engineering, National Taiwan University of Science and Technology, 43, Sec. 4, Keelung Road, Taipei 106, Taiwan; 4Department of Mechanical Engineering, National Yang Ming Chiao Tung University, No. 1001, Daxue Rd. East Dist., Hsinchu City 300093, Taiwan; 5Department of Chemical Engineering, i-Center for Advanced Science and Technology (iCAST), National Chung Hsing University, Taichung 40227, Taiwan

**Keywords:** silver clusters, poly(methacrylic acid) brush, plague, reflectance spectra

## Abstract

A hole array was fabricated via photolithography to wet the bottoms of holes using oxygen plasma. Amide-terminated silane, a water immiscible compound before hydrolysis, was evaporated for deposition on the plasma-treated hole template surface. The silane compound was hydrolyzed along the edges of circular sides of the hole bottom to form a ring of an initiator after halogenation. Poly(methacrylic acid) (PMAA) was grafted from the ring of the initiator to attract Ag clusters (AgCs) as AgC–PMAA hybrid ring (SPHR) arrays via alternate phase transition cycles. The SPHR arrays were modified with a Yersinia pestis antibody (abY) to detect the antigen of Yersinia pestis (agY) for plague diagnosis. The binding of the agY onto the abY-anchored SPHR array resulted in a geometrical change from a ring to a two-humped structure. The reflectance spectra could be used to analyze the AgC attachment and the agY binding onto the abY-anchored SPHR array. The linear range between the wavelength shift and agY concentration from 30 to 270 pg mL^−1^ was established to obtain the detection limit of ~12.3 pg mL^−1^. Our proposed method provides a novel pathway to efficiently fabricate a ring array with a scale of less than 100 nm, which demonstrates excellent performance in preclinical trials.

## 1. Introduction

The optical properties of metal nanostructures have attracted considerable attention for both basic and applied research owing to surface plasmon polaritons (SPPs) [1,2]. The spatial pattern of the electric field on a nanostructured metal surface is crucial in developing metal gratings, which facilitate the optical response of SPPs in nanostructures [3]. The fabrication of inorganic or organic materials with an ordered array has attracted significant attention for various applications, such as surface-enhanced Raman scattering (SERS) [4], photocatalysis [5], heavy-ion sensing [6], bio-architecture [7], and cancer diagnosis [8]. Several types of nanostructures, including nanoparticles [9], nanotubes [10], nanodisks [11], and nanowires [12], have been fabricated with various scales and dimensions to evaluate their specific properties. However, only limited articles have reported the optical properties of these nanostructures with regular orders. In particular, regular-order nanomaterials facilitate the confinement of light to subwavelength dimensions in the form of SPPs owing to their collective oscillations of free electrons around the metal–dielectric interface [13].

In previous studies, nanoring arrays have been widely developed to investigate their fascinating properties in immunoassay [14], cancer-cell counting [15], refractometric devices [16], ATP detection [17], and electrochemistry [18] because of the lightning-rod phenomenon. The lengths, regularities, and diameters of the nanoring are crucial parameters influencing the optical properties. Thus, designable nanoring structures are desirable to precisely modulate the structural parameters of nanorings for optimization. Additionally, the facile fabrication of designable nanorings over a large area with high regularity is another topic for device design and miniaturization in practice. General lithography is commonly used for preparing designable nanorings. However, advanced photolithography and semiconductor processes are needed to fabricate dense nanorings with diameters less than 100 nm [19]. Alternative methods to prepare nanorings less than 100 nm in diameter have also been proposed; these include nanosphere lithography [20], template assistance [21], 3D printing [22], and self-assembly [23]. However, the cost-effectiveness and throughput of these methods are inadequate for large-scale industrial production.

In polymer brushes, one end of the polymer chain is tethered to a substrate, which can modulate the chemical and physical properties of the surface through the functional groups and molecular weights. The functional groups of polymer brushes with electrostatic charges can be used to adsorb the charged metal nanoparticles [24]. For example, gold nanoparticles (AuNPs) have been assembled as a regular disk array on a substrate with a well-defined polymer brush pattern [25,26]. Functional groups of polymer chains can be modulated flexibly with the environmental conditions to attract metal particles based on the affinity or electrostatic forces [27]. Responsive polymer-based materials can alter their affinity and/or electrostatic charges under various external stimuli, and hence, they are called stimuli-responsive polymers [28,29]. Poly(methacrylic acid) (PMAA) is a pH/temperature dual-responsive polymer that undergoes a phase transition in response to the solution pH and temperature. Meanwhile, the affinity, electrostatic charges, surface activity, chain conformation, solubility, and configuration of PMAA can vary upon exposure to external stimuli. The structural and property changes in PMAA with phase transition occur in the temperature range of 21–25 °C, the so-called lower critical solution temperature (LCST), depending on the molecular weights and concentration in 1,2-dimethoxyethane (DME) [30]. The reversible phase transition with property changes in PMAA facilitate attraction of metal salts with negative charges [31]. Silver clusters (AgCs) have unique intrinsic optical and electronic features, as well as physicochemical characteristics, which make them highly significant in modern technology. These intrinsic features of AgCs can be varied by assembling AgCs as spatial arrays through polymer brushes to change the shape, size, and aspect ratio. This assembly of polymer/AgCs composites prepared through the bridging of thiol groups exhibits several properties, such as conductivity and plasmon resonance, by modulation of the distances among AgCs. Thus, the fabrication of well-defined polymer brush patterns is crucial in forming designable polymer/AgCs composite arrays with high regularity over a large area on the substrates [32].

Reflection resonance is an important optical property of periodic arrays of metallic particles, which can enhance the reflection intensity of scattering by particles under appropriate conditions [33]. This study used regular PMAA brush to form a ring structure via photolithography and oxygen plasma processes. A template of hole array was employed to create an alternate wetting area on the plasma-treated bottoms of rings and dewetting area with photoresist cover on the surface, respectively. The as-prepared sample was modified with hydrophobic 3-(ethoxydimethylsilyl)propylamine (3EP) on all wetting and dewetting regions [34]. After removing the template with acetone, the surface appeared ring structure of the hydrolyzed 3EP assembly at the interface between the wetting and dewetting regions. The hydrolyzed 3EP assembly could be modified with halogen groups to generate a ring initiator for synthesis of PMAA brush via atom transfer radical polymerization (ATRP). The ring PMAA brush attracted AgCs, synthesized by a two-phase system, to form the silver clusters/poly(methacrylic acid) hybrid ring (SPHR) array through alternate phase transition processes in DME. Plague is a common infectious disease caused by *Yersinia pestis* (*Y. pestis*). In this regard, the rapid and precise detection of the *Y. pestis* antigen (agY) is expected to help hinder plague spreading. The SPHR arrays were treated with protein G to bind the Y. pestis antibody (abY) for detecting the agY. The ring structure of abY-modified SPHR resulted in optical property changes in these arrays in the reflectance wavelength after agY treatment [35]. Moreover, the inactivated *Y. pestis* was spiked in the blood specimen for preclinical trials. Owing to their excellent performance, the proposed SPHR sensors could be potential candidates for plague diagnosis devices.

## 2. Experimental Section

### 2.1. Materials

Silicon wafers were obtained from Hitachi to pattern the hole template via lithography with an i-line stepper (365 nm, Overlay Accuracy ≤ 45nm, FPA-3000 i5^+^). The compounds 3-(ethoxydimethylsilyl)propylamine (3EP) and 2-bromo-2-methylpropionyl bromide (2B), triethylamine (TA), 1,1,4,7,7-pentamethyldiethylenetriamin (PDA), copper(I) bromide, and copper(II) bromide were purchased from Acros Organics Co. The compound methacrylic acid was obtained from Sigma Aldrich, and the inhibitors were removed before use. Silver perchlorate (AgClO_4_), tetra-*n*-octylammonium bromide, sodium borohydride, *n*-(3-Dimethylaminopropyl)-*N*’-ethylcarbodiimide hydrochloride (EDC), and *N*-hydroxysuccinimide (NHS) were also purchased from Sigma Aldrich for AgC synthesis. Bovine serum albumin (BSA) and AffiniPure goat anti-rabbit IgG (AGRI) were supplied by Taivital Biopharmaceutical Co. Protein G (proG) was obtained from Jackson ImmunoResearch Lab. Purified antibody (abY), antigen (agY) of *Y. pestis*, and inactivated *Y. pestis* were provided by the Institute of Preventive Medicine, National Defense Medical Center, Taiwan, R.O.C. *Escherichia coli* (*E. coli*) and *Staphylococcus aureus* (*S. aureus*) used as negative controls were provided by National Defense Medical Center as well.

### 2.2. Fabrication of the SPHR-Based Biosensor

Figure 1 displays the fabrication strategy of the ring structured SPHR as biosensors on the silicon substrate [36]. A: A blank silicon wafer was treated with hexamethyldisilazane to enhance the affinity between the photoresist and the wafer. A template of the hole array was formed on the wafer surface via lithography. B: The bottom areas of the hole templates were treated with oxygen plasma to enhance their hydrophilicity. C: Water immiscible 3EP was evaporated to deposit on the template surface before hydrolysis. Hydrophobic 3EP was predominately deposited on the template because of the lower affinity with the plasma-treated surface [37]. D: Meanwhile, 3EP was hydrolyzed to assemble along the edge of two sides of the hole bottom to form a ring structure after removing the template. E: The ring halogenation of 3EP assembly by 2B formed the ring initiator for grafting polymerization.

F: PMAA brush rings were formed for 24 h grafting polymerization at 25 °C with the halogen-terminated ring initiator via ATRP in ethanol. G: Sequentially, AgCs, ranging from 9 to 11 nm in the suspension, were synthesized by a two-phase system prior to use (Appendix A) [38]. AgCs were attracted by the PMAA brush rings via a “breathing-in” process of alternate phase transition [39]. The PMAA rings were immersed in the AgCs colloid solution to be heated over the LCST (50 °C) and then reversibly cooled down below the LCST (0 °C). H: AgCs were adsorbed by the ring array of PMAA via the alternate phase transition cycles and, thus, obtained the SPHR that was reacted with proG at a 0.6 mM concentration to immobilize abY on the SPHR with orientation via the EDC/NHS reaction [40]. Coupling agY with the abY-anchored SPHR changed the ring structure, shifting the reflectance peak measured using UV–visible–near-IR spectroscopy (JASCO: V-570).

### 2.3. Optical Properties of the SPHR-Based Biosensors

Reflectance measurements can be used to analyze specific modes of electromagnetic coupling within the interval of a ring structure, depending on its size. The structural transformation from ring to concaves led to a blue shift in the reflectance peak wavelength. Notably, the SPP of metal nanostructures depends on the spatial pattern of the electric field on the nanostructured metal surface [41]. Therefore, various spatial patterns could be monitored by reflectance spectra. AgC density within the SPHR can also be identified by the shift in the reflectance peak wavelength after the alternate phase transition. Additionally, the reflectance peak wavelength shift of our SPHR is expected to be an indicator for biological detection.

The abY-immobilized SPHR (~0.25 cm^2^) were coated with the agY solutions from 0 to 440 pg mL^−1^ concentrations at room temperature for 30 min. After washing the abY-immobilized SPHR with a phosphate buffered saline (PBS) solution, the reflectance spectra of the samples were recorded. BSA and AGRI solutions were also prepared to evaluate the non-specific adsorption on the SPHR array under the same conditions. Furthermore, 90, 150, and 210 pg mL^−1^ agY solutions were added to BSA or AGRI solutions at 10, 100, and 1000 times agY concentrations to evaluate the selectivity performance at room temperature. Three samples of abY-immobilized SPHR were prepared to evaluate the reproducibility, and each data point was recorded as the mean ± standard deviation.

To decrease the risk posed by clinical trials, we used inactivated *Y. pestis* to prepare relevant preclinical samples. The diagnosis of *Y. pestis* infection to help prevent the widespread transmission of plague is commonly performed using blood from rats or wild rodents. The protocol for diagnosing *Y. pestis* infection by the abY-grafted SPHR was as follows. Heparin-containing whole blood (1 mL) obtained from healthy subjects of the Chi Mei Medical Center was spiked with inactivated *Y. pestis* to concentrations of 1–1000 CFU mL^−1^. Each blood specimen (1 mL) was mixed with SDS-PAGE sample buffer (3 mL) to lyse the bacterial cells [42]. After cell lysis, a 100 μL sample aliquot was coated onto a given area (~0.25 cm^2^) of abY-grafted SPHR, incubated for 30 min, washed to remove loose matter, and blown dry with nitrogen gas. LSPR spectra were recorded to analyze the production of agY by *Y. pestis* on the chips and determine agY concentration. *E. coli* and *S. aureus* were employed as controls to investigate the selectivity of the SPHR-based sensor using the same protocol. Three *Y. pestis*-containing blood specimens (100, 200, and 300 CFU mL^−1^) were measured at least thrice, and the data were compared with those obtained using a commercial enzyme-linked immunosorbent assay (ELISA) kit to calculate the inaccuracies of our sensors and record as means ± standard deviations [43]. Furthermore, the competitive conditions of a trinary mixture system, including *Y. pestis*, *E. coli*, and *S. aureus,* were employed to evaluate interferences during detection at 25 °C. The agY samples with various concentrations were also employed to post-spike trinary-mixture samples diluted two- and four-fold with PBS (10 mM, pH 7.4) to calculate the recoveries and bias of the SPHR sensors using the Wilcoxon signed-rank test.

## 3. Results and Discussion

### 3.1. Surface Morphology Analyses at Each Stage for SPHR Fabrication

PMAA, AgCs–PMAA, and abY-grafted AgCs–PMAA layers without ring patterns were analyzed by XPS to avoid the weakening of the ring matter signal by the predominant Si signal of the substrate. Figure 1a displays the wide-scan XPS survey spectra of the above three samples, demonstrating the presence of Si 2s, Si 2p, C 1s, N 1s, and O 1s peaks at 150–153, 99–104, 282–292, 396–403, and 528–535 eV, respectively. Weak N 1s and Br 3d5 peaks in the ranges of 285–293 and 65–73 eV, respectively, were attributed to the halogen-terminated initiator in the chain ends [44]. Except the signals of the PMAA graft, Ag 3d peaks at 365–375 eV appeared in the spectrum after the “breathing-in” process of alternate phase transition for 12 cycles, which indicated the rich AgCs attachment onto PMAA chains [45]. The XPS profile of the abY-grafted SPHR retained the Ag 3d, Si 2s, Si 2p, C 1s, N 1s, and O 1s peaks. Table 1 summarizes the surface compositions of PMAA, SPHR, and abY-grafted SPHR, revealing that the contents of C and O on the PMAA surface (54.91 and 26.51 at%, respectively) agreed with the theoretical C/O molar ratio of two. Notably, Si content significantly decreased after the attachment of AgCs on PMAA brushes, which was ascribed to the shielding of Si by the AgCs-PMAA layer. As the Si-based surface contained Si, C, O, N, and Ag, we used [C]/[Si], [O]/[Si], [N]/[Si], and [Ag]/[Si] ratios to express chemical composition changes. The grafting of abY increased [C]/[Si], [O]/[Si], and [N]/[Si] ratios but slightly decreased the [Ag]/[Si] ratio. Figure 1b–d show the high-resolution C 1s spectra of PMAA, AgCs–PMAA, and abY-grafted AgCs–PMAA. The spectrum of PMAA could be deconvoluted into three peaks at 284.3, 286.2, and 288.6 eV assigned to C–C/H, C–O, and O–C=O bonds, respectively, in line with a previous study [44]. In addition to featuring the three above peaks, the high-resolution C 1s spectrum of AgCs–PMAA exhibited a peak at 286.4 eV attributed to the C–O/C–N bonds of TOAB on AgCs. Modification with proG and abY resulted in the emergence of peaks at 284.3, 285.7, 286.2, 287.4, and 288.6 eV ascribed to the C–C/H, C–N, C–O, N–C=O, and O–C=O bonds of abY-grafted AgCs–PMAA, respectively.

Figure 2a shows the surface morphology of the hole template using AFM. The scale and duty ratio of the photomask were designed as 700 nm and 1.5, respectively. The resulting hole depth and width were ~765 and ~683 nm, respectively, with high regularity similar to our design. The treatment with oxygen plasma led to alternate wetting and dewetting areas at the template bottom. After oxygen plasma treatment, the sample was treated with 3EP vapors and hydrophobic silane before hydrolysis. Meanwhile, the 3EP along the two sides of the hole bottom was hydrolyzed to assemble at the interfaces of the wetting and dewetting areas. The 3EP assembled along the edge of bottoms to form rectangle rings on the substrate after removing the template, as shown in Figure 2b. The scale of the ring was ~703 nm, consistent with the single hole scale. The results verify that 3EP was only selectively hydrolyzed at the interface along the edge of the single hole to form the regular ring pattern with thicknesses ranging from 12–13 nm. Dimensions and structure of the 3EP ring assembly did not change significantly after modifying halogen groups (2B) as the ATRP initiator. The ring structure remained after PMAA was grafted from the ring initiators via ATRP for 24 h. The PMAA brush ring featured a smooth surface, 416 ± 5 nm height, 364 ± 4 nm thickness, 281 ± 6 nm inner and 958 ± 6 nm outer width, respectively (Figure 2c). The scale extension of the smooth rings was attributed to the collapse of the PMAA grafts.

The inner width the rings of the PMAA brush increased marginally after AgCs were attached to the rings via an alternate phase transition process (Figure 2d). The AgCs assemblies also appeared on the inner bottom of the ring, indicating that the “breathing-in” process of the alternate phase transition substantially guided the attachment of AgCs to the PMAA brush. Modifying proG and abY on the SPHR considerably increased the inner height of the ring (Figure 2e). The increase in inner height without a significant change in other geometrical parameters was attributed to the higher affinity of the ring for soft matter on the surface. Analyte adsorption on the nanostructure could change the geometrical parameters and local dielectric environment, which was reported earlier for biosensing [46]. The abY-anchored SPHR was incubated with agY solutions at 150 pg mL^−1^ to observe the structure change. The ring structure turned into a disk-like structure because agY occupied the inner of the ring structure (Figure 2f). Except for the hydrophobic inners of the ring, other bottom regions remained hydrophilic. The agY assembled predominately within the ring inners, indicating that it possessed a higher affinity toward hydrophobic regions than hydrophilic regions.

Figure 3 shows the top and tilted-view scanning electron microscopy (SEM) images of the PMAA ring array after 3, 6, 9, and 12 cycles of AgC attachment using the “breathing-in” process of alternate phase transition, respectively. Ring width increased with the number of “breathing-in” cycles used to form the ring structure. Few AgCs still attached inside the central area of rings after nine “breathing-in” cycles to form a concave-like structure. Except the AgC deposition upon the central area of rings, other geometrical parameters did not change significantly. The results suggest that PMMA brush provided the sufficient support during the AgC attachment. The SPHR formed after AgC attachment via 12 swelling/shrinking cycles was employed as an optical sensor for agY detection in further experiments.

### 3.2. Measurement of Optical Property of the SPHR-Based Sensor

The Fano resonance is a resonant scattering phenomenon caused by the close coexistence of constructive and destructive interference of waves in a system. Interference between a background and a resonant scattering process leads to reflection (or transmission) peaks and dips with asymmetric ring shape at nearby frequencies [47]. This feature indicates that two scattering amplitudes exist; one gives rise to scattering within a continuum of states, and the other causes the excitation of a discrete state. The Fano resonance asymmetric line shape occurs when the wavelength of the resonant state is located within the wavelength range of the continuum states. Generally, the continuum scattering amplitude responds slightly to wavelength, while the resonant scattering amplitude responds to both magnitude and phase sensitively, generating the asymmetric line shape. A change in the structure’s geometry can provide an extra degree of freedom to generate the Fano resonance [48]. Configuration of the structure to create symmetric and antisymmetric modes in the system also facilitates the investigation of typical analogs of electromagnetically induced transparency in plasmonic or planar metamaterials, resulting in inherent Fano resonance in reflectance [49]. Thus, the reflectance spectra were employed to optimize the AgC attachment during the alternate phase transition. The reflectance spectra were collected from ~20 mm^2^ of measuring area with ~1.2 × 10^7^ rings according to our design. The density of AgCs inside the SPHR varied during alternate phase transition cycles. The periodic structured SPHR arrays reflected light at several wavelengths, which resulted in structural coloration. Reflectance peak position depends on the uniformity of the periodic structure, e.g., ring width or thickness. Figure 4a displays the effects of swelling/shrinking cycles on the reflectance spectra of the SPHR, showing that its uniformity resulted in a progressive peak shift with increasing cycle number. The shape and location of reflectance peaks were unstable for three and six cycles, which was attributed to the fact that the periodic structure of the SPHR array did not form regularly upon AgC attachment under these conditions. After nine cycles of the “breathing-in” process, five distinct reflectance peaks stably appeared at 386 (peak I), 444 (peak II), 545 (peak III), 659 (peak IV), and 862 (peak V) nm, indicating that AgC attachment reached saturation to form a perfect ring structure with high uniformity. These five peaks shifted to higher wavelengths with the increasing number of cycles (Figure 4b), ascribed to the concomitant decrease in the distance between AgCs. The reflectance peak location exhibited an approximately linear increase upon increasing cycle number from 3 to 12 and subsequently plateaued, which indicated the saturation of AgC immobilization.

Upon the coupling of abY with agY, the reflectance peaks of abY-modified SPHR shifted to lower wavelengths. Figure 5 shows the detection performance in the concentration range from 0 to 270 pg mL^−1^ with the abY-grafted SPHR using reflectance spectra, revealing that this treatment induced concentration-dependent peak shifts due to ring structure changes. The positions of peaks I and II did not vary with agY concentration. The distinct shift of peak III from 546 to 535 nm was ascribed to the similarity between the wavelength of incident light and ring diameter. Peak IV shifted from 659 to 631 nm with increasing agY concentration because of the concomitant parameter change in the ring geometry. The peak V also blue-shifted from 866 to 813 nm with agY concentration. Binding the biomolecules to bridge the ring also induced the thermal mobility of AgCs and resulted in shifts in the reflectance spectra [50,51]. Thus, our results indicate that higher sensitivity for biomolecule detection was observed when the scale of the SPHR ring structure was less than the wavelength of incident light.

The reflectance peak shift (*P*_s_) was calculated as
*P*_s_ = *P*_0_ − *P_i_*,(1)
where *P*_0_ and *P_i_* represent the reflectance peak wavelengths of abY-grafted SPHR after agY treatment at concentrations of 0 and *i* pg mL^−1^, respectively. Peaks IV and V were selected to evaluate sensing performance because of their increased blue shifts upon agY treatment.

Figure 6a,b display the *P*_s_ of peaks IV and V, respectively, as a function of agY concentration. The wavelengths of these peaks IV and V of abY-grafted SPHR were 659 and 862 nm after coupling with agY at 0 pg mL^−1^, respectively. The *P*_s_ of peak IV did not significantly change at an agY concentration of 30 pg mL^−1^ and the linear range appeared as the agY concentration increased from 60 to 270 pg mL^−1^ (*P*_s_ = 0.1428[agY] − 5.2324, *R*^2^ = 0.9933). The *P*_s_ of peak V also linearly increased as the agY concentration increased from 0 to 270 pg mL^−1^ (*P*_s_ = 0.1826[agY] − 0.2273, *R*^2^ = 0.968) and significantly exceeded that of peak IV, suggesting the higher sensitivity of agY detection in the former case. However, the lower correlation coefficient of the linear relationship for peak V suggested an irregularity of the ring structure variation. In the presence of both interferents of BSA and AGRI, the average *P*_s_ of peaks IV and V at agY concentrations of 0–270 pg mL^−1^ were below 2 nm, i.e., these proteins did not significantly influence *P*_s_. The limits of detection (LODs; signal/noise = 3) for agY were calculated as 51.3 and 12.3 pg mL^−1^ using linear regression equations for peaks IV and V, respectively.

Figure 7a,b display the *P*_s_ values of peaks IV and V, respectively, for different agY concentrations and BSA/AGRI concentrations various times that of agY. At concentrations 10, 100, and 1000 times that of agY ([agY] = 90, 150, and 210 pg mL^−1^), BSA and AGRI had a negligible effect on the *P*_s_ of peaks IV and V, indicating high selectivity attributable to proG modification. The adsorption sites were occupied mostly by proG, which resulted in anti-biofouling properties and high selectivity.

### 3.3. Preclinical Trials

One of the most sensitive commercial kits (ABICAP) provides a meager *Y. pestis* detection limit of 10^4^ CFU mL^−1 in peripheral whole blood.^ To examine the potential clinical application of our SPHR-based sensor, blood samples were spiked with inactivated *Y. pestis* at 1, 10, 20, 40, 100, 300, 600, and 1000 CFU mL^−1^ for agY quantitation after cell lysis. Figure 8 displays the *P*_s_ of peaks IV and V, respectively, as a function of *Y. pestis* concentration, revealing that the *P*_s_ of peak IV did not significantly change below [*Y. pestis*] = 10 CFU mL^−1^ and linearly increased with increasing *Y. pestis* concentration between 20 and 300 CFU mL^−1^ (Figure 8a). *P*_s_ plateaued at 600–1000 CFU mL^−1^, which indicated that the center of the ring was completely occupied by AgCs. The *P*_s_ of peak V also linearly increased with increasing *Y. pestis* concentration from 10 to 300 CFU mL^−1^ and significantly exceeded that of peak IV, suggesting the higher sensitivity of *Y. pestis* detection in the former case. (Figure 8b) The average *P*_s_ values of peaks IV and V did not significantly change in the presence of *E. coli* and *S. aureus* at 1–1000 CFU mL^−1^, i.e., these bacteria did not interfere with *Y. pestis* detection. The LODs of agY (signal/noise = 3) calculated using linear regression equations for peak IV and V were 20 and 10 CFU mL^−1^, respectively. Moreover, blood samples were spiked with inactivated *Y. pestis* at 100, 200, and 300 CFU mL^−1^ for post-lysis agY quantitation using both ELISA and the SPHR sensor (Table 2). The ELISA results were used as reference values to examine the accuracy of our sensor in terms of relative standard deviations (RSDs). All RSDs were ≤5.79%, indicating the high accuracy and reliability of our method. The Wilcoxon signed-rank test was selected to analyze method accuracy and precision. There was no significant difference (*p* > 0.05) between commercial ELISA and our method for agY quantitation in blood specimens containing 50 CFU mL^−1^ of *Y. pestis*, 10^6^ CFU mL^−1^ of *E. coli*, and 10^6^ CFU mL^−1^ of *S. aureus*. Furthermore, the blood samples were diluted two- or four-fold and post-spiked with agY for recovery tests. The recoveries ranged from 96.21 to 104.98%, and all RSDs were ≤6.34%, which indicated that the spiked agY could be accurately and selectively identified in real complex samples (Table 3). The other biomolecules in the matrix did not markedly interfere with agY quantitation. Table 4 summarizes previously reported methods of agY detection. The sensitivity of our SPHR sensor is not extremely high; however, the sensor scale can be increased by further downscaling to improve the sensitivity. Indeed, the designable abY-grafted SPHR arrays were therefore concluded to be applicable as reflectance biosensors.

## 4. Conclusions

We developed a simple route consisting of photolithography processes and surface modification technologies to form designable ring patterns. The nanoscale ring patterns could be metalized with AgCs clusters through polymer brushes by the alternate phase transition of “breathing-in” cycles to construct ring arrays as sensor devices. The proposed approach facilitates the practice of nanoring fabrication in industry. Our scale-free approach can self-organize other organic, metallic, and ceramic particles as nanorings. A charge-carrying PMAA brush is appropriate as the matrix to attract the metal particles and anchor aptamers for the fabrication of the SPHR-based devices through the “breathing-in” process. Reflectance spectra can provide information on density of the AgCs inside the PMAA matrix during the “breathing-in” process and detect agY through the redshift with high sensitivity. Notably, agY was predominately filled within the inner of the hydrophobic ring structure because biomolecules possessed higher affinity in the hydrophobic region than in the hydrophilic region. The wavelengths of the reflectance peak of the SPHR redshift linearly with agY concentration, indicating a novel label-free approach to probing antigens with high sensitivity in the preclinical trials. Our proposed detection platform, the SPHR array, could be a potential candidate with reflectance peak wavelengths for biosensing applications.

## Data Availability

The data presented in this study are available on request from the corresponding author.

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
