# Peer review of "Designable Poly(methacrylic Acid)/Silver Cluster Ring Arrays as Reflectance Spectroscopy-Based Biosensors for Label-Free Plague Diagnosis"

_polymers, 2023, doi:10.3390/polym15081919_

Round 1

Reviewer 1 Report

This manuscript reports the fabrication of hole array by photolitograpgy. The experimental parts is clearly written as the entire paper. The paper is ready for publication in Polymers. 

Author Response

Thanks for the comments.

Reviewer 2 Report

Chen et al. developed a detection platform for label-free plague diagnosis. The whole route for fabrication of hybrid nanoring arrays was described and surface morphology at each stage and optical properties of the sensor were analyzed. The preclinical trials of the sensor showed high sensitivity and selectivity and excellent performance of the detection platform.

The manuscript is very well written with a huge experimental material that is nicely presented. Conclusions are adequate and are supported by the experimental data and findings. I recommend accepting of the manuscript. A minor point can be considered: I did not find steps I and J in Scheme 1.

Author Response

Step I and J are typos. Step I is included in the step H. Step J is the final step. We have remove the symbol I and J from the text.

Reviewer 3 Report

Mauscript ID_polymers-2305845

Title: Designable poly(methacrylic acid)/silver cluster ring arrays as 1 reflectance spectroscopy-based biosensors for label-free plague 2 diagnosis

Reviewer’s comments:

Starting from a hole array fabricated via lithography, the authors create a hybrid ring nanowire array- Poly(methacrylic acid) (PMAA) AgCs–PMAA hybrid ring. It presents an interesting chemical manipulation of surfaces, that results in a procedure closer to technology for obtaining nano-rings. The SPHR arrays were modified with a Yersinia pestis antibody (abY) to detect the antigen of Yersinia pestis (agY) for plague diagnosis. The obtained results are presented in comparison with similar studies in literature.

  1. Since the abbreviation SPHR is not clearly in its components, please write some words about it.
  2. Some data about the i-line stepper might be interesting
  3. Some small typo mistakes (e.g. at r. 426, the references are numbered twice)

I recommend publish after minor corrections.

Author Response

  1. Silver clusters/poly(methacrylic acid) hybrid ring (SPHR) array is abbreviated as SPHR. We have written more words in the text.
  2. We have included the item (365 nm, Overlay Accuracy ≦ 45nm ,FPA-3000 i5+) behind the i-line stepper in the text. 
  3. Reference [28] and [36] are identical. Reference [36] is revised as "J. Guo, B. Huang, J. Lai, C. Lu, J. Chen, Reversibly photoswitchable gratings prepared from azobenzene-modified tethered poly(methacrylic acid) brush as colored actuator. Sens. Actuators: B. Chem. 2020, 304, 127275."